# Sampling in Constrained Domains with Orthogonal-Space Variational Gradient Descent

**Ruqi Zhang**
Department of Computer Science
Purdue University
ruqiz@purdue.edu

**Qiang Liu**
Department of Computer Science
University of Texas at Austin
lqiang@cs.texas.edu

**Xin T. Tong**
Department of Mathematics
National University of Singapore
mattxin@nus.edu.sg

## Abstract

Sampling methods, as important inference and learning techniques, are typically designed for unconstrained domains. However, constraints are ubiquitous in machine learning problems, such as those on safety, fairness, robustness, and many other properties that must be satisfied to apply sampling results in real-life applications. Enforcing these constraints often leads to implicitly-defined manifolds, making efficient sampling with constraints very challenging. In this paper, we propose a new variational framework with a designed orthogonal-space gradient flow (O-Gradient) for sampling on a manifold $\mathcal{G}_0$ defined by general equality constraints. O-Gradient decomposes the gradient into two parts: one decreases the distance to $\mathcal{G}_0$ and the other decreases the KL divergence in the orthogonal space. While most existing manifold sampling methods require initialization on $\mathcal{G}_0$, O-Gradient does not require such prior knowledge. We prove that O-Gradient converges to the target constrained distribution with rate $\widetilde{O}(1/\text{the number of iterations})$ under mild conditions. Our proof relies on a new Stein characterization of conditional measure which could be of independent interest. We implement O-Gradient through both Langevin dynamics and Stein variational gradient descent and demonstrate its effectiveness in various experiments, including Bayesian deep neural networks.

## 1 Introduction

Sampling methods, such as Markov chain Monte Carlo (MCMC) [1] and Stein variational gradient descent (SVGD) [23, 22], have been widely used for getting samples from or approximating intractable distributions in machine learning (ML) problems, such as estimating Bayesian neural network posteriors [38], generating new images [33], and training energy-based models [15]. While being powerful, most sampling methods usually can only be used in unconstrained domains or some special geometric spaces. This greatly limits the application of sampling to many real-life tasks.

We consider sampling from a distribution $\pi$ with an equality constraint $g(x) = 0$ where $g : \mathbb{R}^d \to \mathbb{R}$ is a general differentiable function. The domain in this case is the level set $\mathcal{G}_0 = \{x \in \mathbb{R}^d : g(x) = 0\}$ which is a submanifold in $\mathbb{R}^d$. We do not require additional information about $\mathcal{G}_0$, such as explicit parameterization or known in-domain points, which is in contrast, often demanded by previous methods [3, 35, 2, 19]. The problem defined above includes many ML applications, such as disease

36th Conference on Neural Information Processing Systems (NeurIPS 2022).

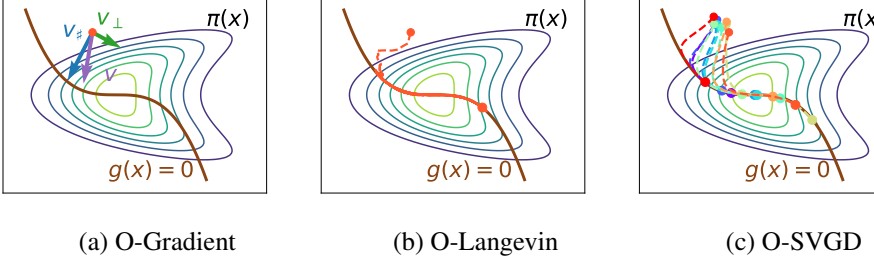

|  (a) O-Gradient | (b) O-Langevin | (c) O-SVGD |

Figure 1: Visualization of our methods. (a) O-Gradient $v$ is formed by $v_\sharp$ which follows $\nabla g$, and $v_\perp$ which is perpendicular to $\nabla g$. (b)-(c) Applying O-Gradient to Langevin dynamics and SVGD. Both methods can approach the manifold and sample on it.

diagnosis with logic rules constraint, policymaking with fairness constraint for different demographic subgroups, and autonomous driving with robustness constraint to unseen scenarios.

In this paper, we propose a new variational framework which transforms the above constrained sampling problem into a constrained functional minimization problem. A special gradient flow, denoted *orthogonal-space gradient flow* (O-Gradient), is developed to minimize the objective. As illustrated in Figure 1a, the direction of O-Gradient $v$ can be decomposed into two parts: the first part $v_\sharp$ drives the sampler towards the manifold $\mathcal{G}_0$ following $\nabla g$ and keeps it on $\mathcal{G}_0$ once arrived; the second part $v_\perp$ makes the sampler explore $\mathcal{G}_0$ following the density $\pi(x)$. We prove the convergence of O-Gradient in the continuous-time mean-field limit. O-Gradient can be applied to both Langevin dynamics and SVGD, resulting in O-Langevin and O-SVGD respectively. As shown in Figure 1b&c, both methods can converge to the target distribution on the manifold. In particular, O-Langevin converges following a noisy trajectory while O-SVGD converges smoothly, similar to their standard unconstrained counterparts. We empirically demonstrate the sampling performance of O-Langevin and O-SVGD across different constrained ML problems. We summarize our contributions as follows:

- We reformulate the hard-constrained sampling problem into a functional optimization problem and derive a special gradient flow, O-Gradient, to obtain the solution.

- We prove that O-Gradient converges to the target constrained distribution with rate $\widetilde{O}(1/\text{the number of iterations})$ under mild conditions. Our proof technique includes a new Stein characterization of conditional measure which could be of independent interest.

- We implement O-Gradient through both Langevin dynamics and SVGD and demonstrate its effectiveness in various experiments, including a constrained synthetic distribution, income classification with fairness constraint, loan classification with logic rules and image classification with robust Bayesian deep neural networks.

## 2   Related Work

**Sampling on Explicitly Defined Manifolds**   Manifolds with special shapes, such as geometric or physics structures, can sometimes be explicitly parameterized in lower dimension spaces. For example, a torus embedded in $\mathbb{R}^3$ can be explicitly defined in two dimensions using polar coordinates. Variants of classical methods have been developed to sample on such manifolds, including rejection sampling [6], Langevin dynamics [35], Hamiltonian Monte Carlo (HMC) [3] and Riemannian manifold HMC [16, 28]. However, explicit parameterization is only applicable to a few special cases and cannot be used for general machine learning problems. In contrast to this line of work, our method is able to work with more general manifolds defined in the original domain $\mathbb{R}^d$.

**Sampling on Implicitly Defined Manifolds**   Many common applications are not endowed with simple manifolds, such as molecular dynamics [17], matrix factorization [27] and free energy calculations [34]. Motivated by these applications, sampling methods on implicitly defined manifolds have been developed. Brubaker et al. [2] has proposed a family of constrained MCMC methods by applying Lagrangian mechanics to Hamiltonian dynamics. Zappa et al. [37] has introduced a constrained Metropolis-Hastings (MH) with a reverse projection check to ensure the reversibility.

Later, this method has been extended to HMC [19] and multiple projections [20]. However, the implementation and analysis of these methods often assume the algorithm starts on the manifold and never leaves it, requiring prior known points on the manifold and expensive projection subroutines, such as Newton's method [2, 37, 18–20] or a long time ordinary differential equation (ODE) [39, 31, 14]. In contrast, our method works with distributions supported on the ambient space and thus gets rid of the above strong assumptions, leading to a much faster update per iteration. This makes our method especially suitable for complex ML models such as deep neural networks.

**Sampling with a Moment Constraint**   Recently, sampling with a general moment constraint, such as $\mathbb{E}_q[g] \leq 0$ where $q$ is the approximated distribution, has been studied [25]. However, this type of constraint can not guarantee every sample to satisfy $g(x) = 0$. From a technical view, the target distribution with a moment constraint is usually not singular w.r.t. $\pi$, so the problem is conceptually less challenging compared to the problem considered in this work.

## 3   Preliminaries

**Variational Framework**   We review the derivation of Langevin dynamics and SVGD from a unified variational framework. The variational approach frames the problem of sampling into a KL divergence minimization problem: $\min_{q \in \mathcal{P}} \text{KL}(q \,\|\, \pi)$ where $\mathcal{P}$ is the space of probability measures. We start from an initial distribution $q_0$ and an initial point $x_0 \sim q_0$, and update $x_t$ following $dx_t = v_t(x_t)dt$, where $v_t \colon \mathbb{R}^d \to \mathbb{R}^d$ is a velocity field at time $t$. Then the density $q_t$ of $x_t$ follows Fokker-Planck equation: $dq_t/dt = -\nabla \cdot (v_t q_t)$, and the KL divergence decreases with the following rate [23]:

$$-\frac{d}{dt}\text{KL}(q_t \,\|\, \pi) = \mathbb{E}_{q_t}[\mathcal{A}_\pi v_t] = \mathbb{E}_{q_t}[(s_\pi - s_{q_t})^\top v_t], \tag{1}$$

where $\mathcal{A}_\pi v(x) = s_\pi(x)^\top v(x) + \nabla \cdot v(x)$ is the Stein operator, and $s_p = \nabla \log p$ is the score function of the distribution $p$. The optimal $v_t$ is obtained by solving an optimization in a Hilbert space $\mathcal{H}$,

$$\max_{v \in \mathcal{H}} \mathbb{E}_{q_t}[(s_\pi - s_{q_t})^\top v] - \frac{1}{2}\|v\|_\mathcal{H}^2. \tag{2}$$

The above objective makes sure that $v_t$ decreases the KL divergence as fast as possible.

**Langevin Dynamics and SVGD Algorithms**   Both Langevin dynamics and SVGD can be derived from this variational framework by taking $\mathcal{H}$ to be different spaces. Taking $\mathcal{H}$ to be $\mathcal{L}_q^2$, the velocity field becomes $v_t(\cdot) = \nabla s_\pi(\cdot) - \nabla q_t(\cdot)$ which can be simulated by Langevin dynamics $dx_t = s_\pi(x_t)dt + dW_t$ with $W_t$ being a standard Brownian motion. After discretization with a step size $\eta > 0$, the update step of Langevin dynamics is $x_{t+1} = x_t + \text{Langevin}(x_t)$, where

$$\text{Langevin}(x_t) = \eta \nabla \log \pi(x_t) + \sqrt{2\eta}\xi_t, \ \ \xi_t \sim \mathcal{N}(0, I). \tag{3}$$

Taking $\mathcal{H}$ to be the reproducing kernel Hilbert space (RKHS) of a continuously differentiable kernel $k \colon \mathbb{R}^d \times \mathbb{R}^d \to \mathbb{R}$, the velocity field becomes $v_t(\cdot) = \mathbb{E}_{x \sim q_t}[k_t(\cdot, x)s_\pi(x) + \nabla_x k_t(\cdot, x)]$. After discretization, the update step of SVGD for particles $\{x_i\}_{i=1}^n$ is $x_{i,t+1} = x_{i,t} + \eta \cdot \text{SVGD}_k(x_{i,t})$, for $i = 1, \ldots, n$, where $\eta$ is a step size and

$$\text{SVGD}_k(x_{i,t}) = \frac{1}{n}\sum_{j=1}^n k(x_{i,t}, x_{j,t})\nabla_{x_{j,t}} \log \pi(x_{j,t}) + \nabla_{x_{j,t}} k(x_{i,t}, x_{j,t}). \tag{4}$$

## 4   Main Method

In this section, we formulate the constrained sampling problem into a constrained optimization through the variational lens in Section 4.1, and introduce a new gradient flow to solve the problem in Section 4.2. We apply this general framework to Langevin dynamics and SVGD, leading to two practical algorithms in Section 4.3.

### 4.1   Constrained Variational Optimization

Recall that our goal is to draw samples according to the probability of $\pi$, but restricted to a low dimensional manifold specified by an equality: $\mathcal{G}_0 := \{x \in \mathbb{R}^d \colon g(x) = 0\}$. Similar to the standard

variational framework in Section 3, we can formulate the problem into a constrained optimization in the space of probability measures:

$$\min_{q \in \mathcal{P}} \mathrm{KL}(q \,||\, \pi), \quad \text{s.t.} \quad q(g(x) = 0) = 1.$$

However, this problem is in general ill-posed. To see that, when $q$ satisfies the constraint, $q$ will be singular w.r.t. $\pi$, so both $\frac{dq}{d\pi}$ and $\mathrm{KL}(q \,||\, \pi)$ are not defined. Although the problem is ill-posed, we are actually still able to derive a KL-gradient flow to solve the problem by considering $q$ supported on $\mathbb{R}^d$. The intuition of the derivation is that, in addition to minimizing the objective as in Eq. (1), the velocity filed $v_t$ should also push $q$ towards $\mathcal{G}_0$ to satisfy the constraint. Surprisingly, the distribution $q_t$ following such a gradient flow indeed converges to the target distribution on the manifold. We will focus on the derivation of the gradient flow here and leave its rigorous justification in Section 5.

## 4.2 Orthogonal-Space Gradient Flow (O-Gradient)

As mentioned above, besides maximizing the decay of $\mathrm{KL}(q \,||\, \pi)$, the velocity field $v_t$ also needs to drive $q$ towards the manifold satisfying $g(x) = 0$. In particular, we add to (2) a requirement that the value of $g(x)$ is driven towards 0 with a given rate:

$$v_t = \arg \max_{v \in \mathcal{H}} \mathbb{E}_{q_t}[(s_\pi - s_{q_t})^\top v] - \frac{1}{2} \|v\|_{\mathcal{H}}^2, \quad \text{s.t.} \quad v_t(x)^\top \nabla g(x) = -\psi(g(x)) \tag{5}$$

where $\psi(x)$ is an increasing odd function. To see the effect of the constraint term, we consider three cases:

- When $g(x) > 0$, then $v_t(x)^\top \nabla g(x) = -\psi(g(x)) > 0$ which ensures that $v_t$ will make $g$ decrease strictly.

- When $g(x) < 0$, then $v_t(x)^\top \nabla g(x) = -\psi(g(x)) < 0$ which ensures that $v_t$ will make $g$ increase strictly.

- When $g(x) = 0$, then $v_t(x)^\top \nabla g(x) = -\psi(g(x)) = 0$ which ensures $x$ stay on the manifold $\mathcal{G}_0$.

We choose $\psi(x) = \alpha \mathrm{sign}(x) |x|^{1+\beta}$ with $\alpha > 0$ and $\beta \in (0, 1]$ in this paper because it is one of the simplest functions that satisfy the requirements and we found it works well in theory and practice.

In summary, the objective function in Eq. (5) is the same as Eq. (2) in the standard variational framework while the constraint ensures that $v_t$ pushes $q_t$ towards the manifold and keeps it stay. It is easy to see that the solution of the above problem can be decomposed as $v_t = v_\sharp + v_\perp$ where

$$v_\sharp(x) = \frac{-\psi(g(x))\nabla g(x)}{\|\nabla g(x)\|^2}, \quad v_\perp \perp \nabla g. \tag{6}$$

We use $f \perp g$ to denote (pointwise) orthogonality: $f(x)^\top g(x) = 0, \forall x \in \mathbb{R}^d$. Note that $v_\sharp$ is parallel to $\nabla g$ and the remaining is to determine $v_\perp$. Note that $v_\perp$ can be represented as a projection of an arbitrary function $u$ to the orthogonal space of $\nabla g$:

$$v_\perp = D(x)u(x), \quad \text{where } D(x) := I - \frac{\nabla g(x)\nabla g(x)^\top}{\|\nabla g(x)\|^2}. \tag{7}$$

The projection operator $D$ makes sure that $v_\perp \perp \nabla g$ holds for any $u$, of which the optimal value we can get by maximizing the unconstrained objective in Eq. (2),

$$\max_{v_\perp} E_{q_t}[(s_\pi - s_{q_t})^\top (v_\sharp + v_\perp)] - \frac{1}{2} \|v_\perp\|_{\mathcal{H}}^2 \Rightarrow \max_u \mathbb{E}_{q_t}[(D(s_\pi - s_{q_t}))^\top u] - \frac{1}{2} \|Du\|_{\mathcal{H}}^2. \tag{8}$$

The optimal solution of $u$ depends on the choice of space $\mathcal{H}$, which we discuss in Section 4.3.

Overall, we obtain the velocity field $v_t$ by first formulating a constrained optimization and then transforming it into an unconstrained optimization via orthogonal decomposition. We call $v_t$ *Orthogonal-Space Gradient Flow* (O-Gradient) and it drives $q_t$ to the target distribution only with the knowledge of $\nabla g$, requiring no explicit representation of the manifold $\mathcal{G}_0$.

## 4.3 Practical Algorithms

After deriving O-Gradient for general Hilbert spaces $\mathcal{H}$, we explain how to implement it using SVGD and Langevin dynamics. The resulting O-SVGD and O-Langevin are outlined in Algorithm 1. At a high level, our algorithms keep the original SVGD or Langevin dynamics movement in the directions perpendicular to $\nabla g$, while pushing the density towards $\mathcal{G}_0$ along the $\nabla g$ direction.

**O-SVGD**  We apply O-Gradient to SVGD first since it is fairly straightforward. Recall that $v_\sharp$ can be obtained using Eq. (6). We solve Eq. (8) to get $v_\perp$ through the following lemma.

**Lemma 4.1.** *When $\mathcal{H}$ is an RKHS with kernel $k \colon \mathbb{R}^d \times \mathbb{R}^d \to \mathbb{R}$, a solution to Eq. (8) is $v_\perp = Du(x) = \mathbb{E}_{y \sim q_t}(k_\perp(x, y) s_\pi(y) + \nabla_y \cdot k_\perp(x, y))$ with the orthogonal-space kernel $k_\perp(x, y) = k(x, y)D(x)D(y)$. Here $k_\perp \colon \mathbb{R}^d \times \mathbb{R}^d \to \mathbb{R}^{d \times d}$ is matrix valued, and $\nabla_y \cdot k_\perp = \sum_j \partial_{y_j} k_\perp^{ij}(x, y)$.*

Then the combined velocity is obtained using the original SVGD with the kernel $k_\perp$,

$$v_t(x) = v_\sharp(x) + \int k_\perp(x, y) s_\pi(y) q_t(y) dy + \int \nabla_y \cdot (k_\perp(x, y)) q_t(y) dy.$$

Numerically, we iteratively update a set of $n$ particles $\{x_{i,t}\}_{i=1}^n \subset \mathbb{R}^d$, such that its empirical distribution $\sum_{i=1}^n \delta_{\theta_{i,t}}/n$ is an approximation of $q_t$ in a proper sense when step size $\eta \to 0$ and particle size $n \to +\infty$. Similar to the update of standard SVGD in Eq. (4), the update of O-SVGD is $x_{i,t+1} = x_{i,t} + \eta \cdot (v_\sharp(x_{i,t}) + \text{SVGD}_{k_\perp}(x_{i,t}))$ where

$$\text{SVGD}_{k_\perp}(x_{i,t}) = \frac{1}{n} \sum_{j=1}^n k_\perp(x_{i,t}, x_{j,t}) \nabla_{x_{j,t}} \log \pi(x_{j,t}) + \nabla_{x_{j,t}} k_\perp(x_{i,t}, x_{j,t}). \tag{9}$$

It is worth noting that $\text{SVGD}_{k_\perp}$ is identical to Eq. 4 but with kernel $k_\perp$ rather than $k$.

**O-Langevin**  The Langevin implementation requires some additional derivation. First of all, with $\mathcal{H} = L_q^2$, we can show that the optimal velocity field is $v_t(x) = \phi(x) - D(x)s_{q_t}(x)$ where $\phi(x) = v_\sharp(x) + D(x)s_\pi(x)$. This leads to a density flow

$$\frac{d}{dt} q_t(x) = -\nabla \cdot (\phi(x) q_t(x)) + \nabla \cdot (D(x) \nabla q_t(x)). \tag{10}$$

Next, we try to design a stochastic differential equation (SDE) of which the Fokker–Plank equation (FPE) is identical to Eq. (10). The result is given by the following:

**Theorem 4.2.** *Consider a vector field $r(x) = \nabla \cdot D(x)$, or its component-wise formulation $r_i(x) = \sum_{j=1}^d \partial_{x_j} D_{i,j}(x)$, where $x_j$ denotes the $j$th dimension of $x$. Consider the SDE*

$$dx_t = (\phi(x_t) + r(x_t))dt + \sqrt{2}D(x_t)dW_t \tag{11}$$

*with $\phi(x) = v_\sharp(x) + D(x)s_\pi(x)$, then its FPE is identical to Eq. (10). Moreover, i) the value $g(x_t)$ has deterministic decay $\frac{d}{dt} g(x_t) = -\psi(x_t)$; ii) for any $f$ with $\nabla f \perp \nabla g = 0$, the generator of $x_t$ satisfies the Langevin equation: $\frac{d}{dt}\mathbb{E}[f(x_t)|x_0 = x]\big|_{t=0} := \mathcal{L}f(x) = \nabla f^\top(x)s_\pi(x) + \Delta f(x)$.*

It is worth pointing out that if $g(x_0) = 0$, then $g(x_t) \equiv 0$, that is, $x_t$ always stays on $\mathcal{G}_0$. In this case, Eq. (11) degenerates to manifold Langevin dynamics studied in previous work [9, 35]. However, our SDE does not have this requirement since *it is still well-defined off $\mathcal{G}_0$*. This is especially useful for numerical implementations, leading to a fast algorithm without expensive projection steps.

Similar to the standard Langevin dynamics update in Eq. (3), the update rule of O-Langevin is $x_{t+1} = x_t + \eta \cdot v_\sharp(x_t) + \text{Langevin}_\perp(x_t)$ where

$$\text{Langevin}_\perp(x_t) = \eta D(x_t)s_\pi(x_t) + \eta r(x_t) + \sqrt{2\eta}D(x_t)\xi_t, \ \ \xi_t \sim \mathcal{N}(0, I). \tag{12}$$

## 5 Theoretical Analysis

We theoretically justify the convergence of O-Gradient in this section. To do so, we first describe the target measure as a conditioned measure $\Pi_0$, then derive its associated orthogonal-space Fisher divergence, and finally prove that O-Gradient converges to $\Pi_0$.

---

**Algorithm 1** O-SVGD and O-Langevin.

> **given:** Initialization $\{x_{i,0}\}_{i=1}^n$ for O-SVGD and $x_0$ for O-Langevin, step size $\eta$.
> **loop**
>     **compute** $v_\sharp(x) = \frac{-\psi(g(x))\nabla g(x)}{\|\nabla g(x)\|^2}$ and the projection operator $D(x) = I - \frac{\nabla g(x)\nabla g(x)^\top}{\|\nabla g(x)\|^2}$.
>     **if** `O-SVGD`, update $\{x_{i,t}\}_{i=1}^n$ by Eq. (9).
>     **if** `O-Langevin`, update $x_t$ by Eq. (12).
> **end loop**

---

### 5.1 Conditioned measure and its Stein characterization

First of all, conditioning on a zero-measure set is a challenging concept. Assume we have a distribution $\Pi$ with density $\pi$. Let $A$ be a set with $\Pi(A) \neq 0$, then $\Pi(B|A) = \frac{\Pi(B \cap A)}{\Pi(A)}$. However, if $\Pi(A) = 0$, this definition is ill-posed. Instead, a standard way to define a conditioned measure on a zero measure set is using disintegration theorem [5]. Let $\Pi_z(\,\cdot\,) = \Pi(\,\cdot\,|g(x) = z)$ be the measure such that

$$\mathbb{E}_\Pi[f(x)] = \mathbb{E}_{z \sim \Pi^g} \mathbb{E}_{x \sim \Pi_z}[f(x)], \quad \forall f, \tag{13}$$

where $\Pi^g$ is the pushforward measure of $\Pi$ under $g$. It is worth noting that $\Pi_0$ *is not* the measure with Hausdorff density $\pi$ on $\mathcal{G}_0$. Instead, under regularity conditions, $\Pi_0$ can also be defined through Hausdorff density $\pi(x)/|\nabla g(x)|$ on $\mathcal{G}_0$, see Lemma 6.4.1 of [32].

One can also avoid this rather abstract definition, and consider an alternative definition which is more intuitive and Bayesian. Suppose we observe $z = g(x) + \sigma\xi$, where $\xi \sim \mathcal{N}(0, 1)$. Under prior $x \sim \pi$, the posterior $\pi_{\sigma^2,z}(x) \propto \pi(x)\exp(-|z - g(x)|^2/2\sigma^2)$. Then we can show that $\Pi_z$ is the (weak) limit of $\pi_{\sigma^2,z}$ as $\sigma \to 0$, moreover, it satisfies a *Stein characterization of conditional measure*. We need some regularity assumptions to state our results.

**Definition 5.1.** *A density $q$ is $g$-regular if there is a constant $L$ such that for small enough $\eta > 0$,*

$$|\mathbb{E}_{\xi \sim \mathcal{N}(0,1)}[q^g(z + \eta\xi)/q^g(z)] - 1| \leq L\eta, \quad \forall z. \tag{14}$$

Note that Eq. (14) holds if $q^g$ is a Lipschitz function. Moreover, while Definition 5.1 requires Eq. (14) to hold for all $z$, in practice we only concern $\Pi_0$, thus only need the equation to hold for $z$ near 0.

**Proposition 5.2.** *Suppose $\pi$ is $g$-regular. Then the weak limit of $\pi_{\eta,z}(x) \propto \pi(x)\exp(-\frac{1}{2\eta}(g(x)-z)^2)$ as $\eta \to 0$ concentrates on $\mathcal{G}_z$ and satisfies Eq. (13). Moreover, for each $z$,*

$$\mathbb{E}_{\Pi_z}[\mathcal{A}_\pi\phi] = 0, \quad \forall\phi \perp \nabla g. \tag{15}$$

The first part of Proposition 5.2 also provides us with a way to sample $\Pi_z$, as we can sample a sequence of distribution $\pi_{\eta_k,z}$ with $\eta_k \to 0$. However, this method involves double loops and is usually much more expensive than single-loop algorithms.

The second part of Proposition 5.2 can be used to formulate an orthogonal-space Stein equation. Specifically, given a $q$ concentrated on $\mathcal{G}_z$, we consider checking whether $\mathbb{E}_q[\mathcal{A}_\pi\phi]$ is zero for $\phi \in \mathcal{H}_\perp = \{\phi : \phi \perp \nabla g\}$, which is a necessary condition for $q = \Pi_z$. Notably, this equation is well defined in $\mathbb{R}^d$ without using any parameterization of $\mathcal{G}_z$.

Compared with the standard Stein equation, we add an additional restriction that $\phi \in \mathcal{H}_\perp$. We argue such restriction is actually very natural. To see that, we note $\Pi_z$ is concentrated on $\mathcal{G}_z$, so we should check the Stein equation with vector field that "live" only on $\mathcal{G}_z$, namely the tangent bundle $\mathcal{T}\mathcal{G}_z$. Notably, if $\phi(x) \in \mathcal{T}\mathcal{G}_z$, then $\phi(x) \perp \nabla g(x)$, therefore it is natural to consider $\mathcal{H}_\perp$.

### 5.2 Orthogonal-Space Fisher divergence

Next we consider how to generalize Fisher divergence for constrained sampling. Since the orthogonal-space Stein equation can only discern discrepancy in $\mathcal{H}_\perp$, it is natural to consider an orthogonal-space Fisher divergence

$$F_\perp(q, \pi) := \|D(s_q - s_\pi)\|_\mathcal{H}^2, \tag{16}$$

where $D$ is the $\mathcal{H}_\perp$-projection operator defined pointwisely using Eq. (7). Its easy to see $F_\perp$ is well defined for a density $q$ on $\mathbb{R}^d$. But it remains unclear whether it can be used to measure the distance between $q$ and $\Pi_z$.

To answer this question, we need some form of strong convexity conditions. Recall that in the unconstrained case, such conditions can be characterized by Poincaré inequalities (PI) when $\mathcal{H}$ is $L_q^2$, so we will focus on this setting. PI counterparts of SVGD remain an open question [10, 7], which we hope to be addressed by future works. When constrained on $\mathcal{G}_z$, PI's formulation involves gradients so it depends on the Riemannian metric used in $\mathcal{G}_z$. One natural choice of Riemannian metric is the metric inherited from $\mathbb{R}^d$ which equals $\|D(x)\nabla f(x)\|$, since $D(x)$ is also the projection onto the tangent bundle of $\mathcal{G}_z$. This leads to the following definition.

**Definition 5.3.** $\Pi_z$ *follows a $\kappa$-PI, if for any smooth $f$ on $\mathbb{R}^d$, $\mathrm{var}_{\Pi_z}[f] \leq \kappa\mathbb{E}_{\Pi_z}[\|D\nabla f\|^2]$.*

PI on Riemannian manifold has been studied in previous work [12, 21]. In particular, Theorem 2.10 of [12] shows that PI holds for the Riemannian measure if the manifold $\mathcal{G}_z$ is compact, and hence it also holds for any equivalent distributions. But it is often hard to find the exact PI constant $\kappa$ for $\Pi_z$, especially if we are interested in a family of $\Pi_z$ with possibly discontinuous dependence on $z$.

**Proposition 5.4.** *Suppose that $\Pi_z$ satisfies $\kappa$-PI for $|z| \leq \delta$, and $q$ is supported on $\{x : |g(x)| \leq \delta\}$. Suppose also that $\Pi_z, q^g, \pi^g$ admit $C^1$ density functions. Then for any function $f$ such that $|f| \leq 1$, the following holds*

$$|\mathbb{E}_q[f] - \mathbb{E}_{\Pi_0}[f]| \leq \sqrt{8\kappa\mathbb{E}_q[\|D(s_q - s_\pi)\|^2]} + \max_{|z| \leq \delta}|\mathbb{E}_{\Pi_z}[f] - \mathbb{E}_{\Pi_0}[f]|.$$

In particular, Proposition 5.4 shows that if we have a $q$ such that it is 1) supported close to $\mathcal{G}_0$ and 2) $F_\perp(q, \pi)$ is small, then $q$ is close to $\Pi_0$ in total variation. We can also interpret the bound alternatively as a decomposition of the difference between $q$ and $\Pi_0$: the first part measures the difference along $\mathcal{H}_\perp$ directions and it is controlled by $F_\perp$; the second part measures the difference along $\nabla g$ direction, and it is controlled by the distance between $q$'s support to $\mathcal{G}_0$.

As a final remark, the orthogonal-space Stein equation (15) and orthogonal-space Fisher divergence (16) can be applied to densities supported on $\mathbb{R}^d$, and their formulations do not require information of $\mathcal{G}_0$ such as parameterization and geodesic. Therefore, they can be used as very computationally friendly statistical divergence in practice.

**Proposition 5.5.** *Suppose we apply $v_t = v_\sharp + Du$ with (6) and (8) to the density field $q_t$.*

1. *Let $M_t = \max\{|g(x)|, x \in supp(q_t)\}$. Let $S_t$ be the solution of an ordinary differential equation $\dot{S}_t = -\psi(S_t)$. Suppose $S_0 = M_0 < \infty$, then $M_t < S_t$ a.s..*

2. *Assume $\psi$ is differentiable with derivative $\dot\psi$. Suppose $g$ is also smooth enough so that*

$$\left|\nabla g^T s_\pi + \Delta g - \frac{2\nabla g^T \nabla^2 g \nabla g}{\|\nabla g\|^2}\right| \leq C_0\|\nabla g\|^2. \tag{17}$$

*Then the KL-divergence follows $\frac{d}{dt}\mathrm{KL}(q_t\|\pi) \leq -F_\perp(q_t, \pi) + \mathbb{E}_{q_t}[|\dot\psi(g)|] + C_0\mathbb{E}_{q_t}[\psi(g)|\|\nabla g\|^2]$.*

The first result shows that the support of $q_t$ will shrink towards $\mathcal{G}_0$. But in order to avoid possible stagnation case, e.g. $\|g\|$ is too small, it is necessary to impose (17). The second result provides us some hint on the choice of $\psi$. Note that under $q_t$ for large enough $t$, $g(x)$ will take values close to 0, so we need $\psi(0) = \dot\psi(0) = 0$. Since $\psi(x)$ should have the same sign with $x$, one natural choice would be $\psi(y) = \alpha\mathrm{sign}(y)|y|^{1+\beta}$.

**Theorem 5.6.** *Suppose we choose $\psi(y) = \alpha sign(y)|y|^{1+\beta}$ for an $\alpha > 0, \beta \in (0, 1]$ and the conditions of Proposition 5.5 hold. Suppose also that $\mathrm{KL}(q_0, \pi) < \infty$ and $\|\nabla g(x)\| \leq D_0$ if $|\psi(x)| < M_0$, then $M_T = O(T^{-\frac{1}{\beta}})$, $\min_{t \leq T} F_\perp(q_t, \pi) = O(\log T/T)$. The orthogonal-space Stein equality holds approximately, in the sense that for any $\phi(x) \perp \nabla g(x)$ and $\|\phi\|_{\mathcal{H}} = 1$, $\min_{t \leq T} |\mathbb{E}_{q_t}\mathcal{A}_\pi\phi| \leq \sqrt{F_\perp(q_t, \pi)} = O(\sqrt{\log T/T})$.*

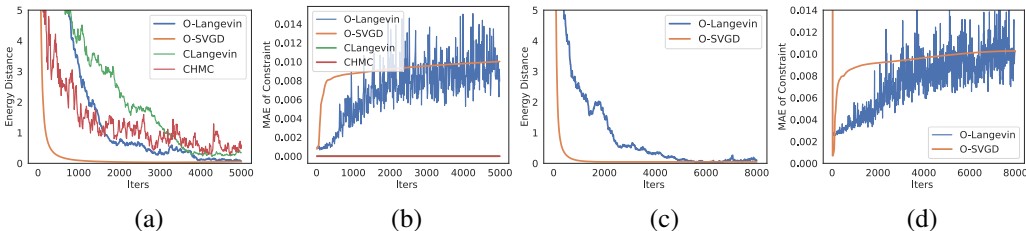

Figure 2: Energy distance and MAE w.r.t. iterations when starting: (a)&(b) on the manifold and (c)&(d) outside the manifold.

Theorem 5.6 shows that we can obtain a $q_t$ that is supported on $\mathcal{G}_{[-\delta,\delta]} = \{x : |g(x)| \leq \delta\}$, while it has close to zero orthogonal-space Fisher divergence and the Stein identity holds approximately on $\mathcal{H}_\perp$. Combining it with Proposition 5.4, we have

**Corollary 5.7.** *Under the conditions of Theorem 5.6, for a bounded $f$ with $\mathcal{H} = L_q^2$, suppose $h_f(z) = \mathbb{E}_\pi[f|g(X) = z] = \mathbb{E}_{\Pi_z}[f]$ is Lipschitz with near $z = 0$, then $\min_{t \leq T} |\mathbb{E}_{q_t}[f] - \mathbb{E}_{\Pi_0}[f]| \leq O(\sqrt{\log T / T})$.*

## 6 Experiments

We demonstrate the effectiveness of our methods on various tasks, including a constrained synthetic distribution, an income classification with a fairness constraint, a loan application with a logic constraint and Bayesian deep neural networks with a robustness constraint. For all MCMC methods, we run $n$ parallel chains and collect the final samples. For all SVGD, we use $n$ particles with an RBF kernel. We released the code at https://github.com/ruqizhang/o-gradient.

**Synthetic Distribution** We first demonstrate our methods on a two-dimensional synthetic distribution where the ground truth samples are available. Let $y \sim \mathcal{N}(0, I)$, and we transform $y$ to $x = [x_1, x_2]$ by $x = \phi^{-1}(y)$ where $\phi(x) = [x_1 + x_2^3, x_2]^\top$. We aim to sample from $\pi(x)$ with constraint $g(x) = x_1 + x_2^3 = 0$. We are able to generate ground truth samples by first generating $y_0 \sim \mathcal{N}(0, 1)$ and then setting $x_{\text{gt}} = [-y_0^3, y_0]$. We report the energy distance between the collected samples and the ground truth samples (measuring how well samples approximate the target distribution) and the mean absolute error (MAE) $|g(x)|$ (measuring how well the samples satisfy the constraint). We set $n = 50$ for this task.

In order to compare with previous manifold sampling methods, which all require initializations on the manifold, we first consider the case when the sampler starts on the manifold. We compare with constrained Langevin dynamics (CLangevin) and constrained Hamiltonian Monte Carlo (CHMC), which are two advanced manifold samplers [19]. From Figure 2a, we observe that O-SVGD and O-Langevin converge after 2000 and 4000 steps respectively whereas CLangevin and CHMC have not fully converged even after 5000 steps. These results indicate that projection-based methods could be inefficient due to finding sub-optimal solutions in the projection subroutines. In terms of runtime, we observe that O-Langevin converges the fastest (see Appendix for comparisons). Figure 2b shows that previous methods have almost zero MAE all the time since they are required to stay on the manifold. Our methods have reasonably small MAE (but not zero due to operating in the ambient space) even without expensive projection steps. O-Langevin has more wiggly trajectories than O-SVGD due to injecting the Gaussian noise.

Next, we test our methods starting outside the manifold, where previous manifold sampling methods are not applicable. From Figure 2d&e, we observe again that O-SVGD converges the fastest in terms of iterations and both methods converge very quickly to the manifold and are able to stay close to it.

**Income Classification with Fairness Constraint** ML Fairness has gained increasing attention in recent years, which aims to guarantee unbiased treatments for individuals w.r.t. gender, race, disabilities, etc. Following previous work [26, 24], we predict whether an individual's annual income is greater than $50,000$ unfavorably in terms of the gender. We use *Adult Income* dataset [13] and train a Bayesian neural network with two-layer multilayer perceptron (MLP). During testing, we use CV score, a standard fairness measure of disparate impact, as the evaluation metric [4]. On this task,

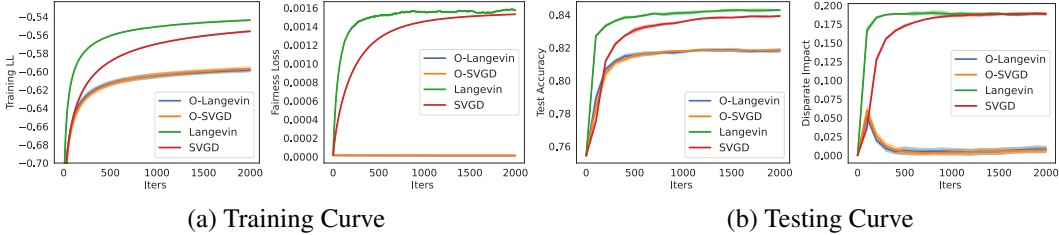

(a) Training Curve                 (b) Testing Curve

Figure 3: Income Classification: O-Langevin and O-SVGD have almost zero fairness loss and disparate impact while maintaining great prediction performance. Standard Langevin and SVGD fail to make fair predictions.

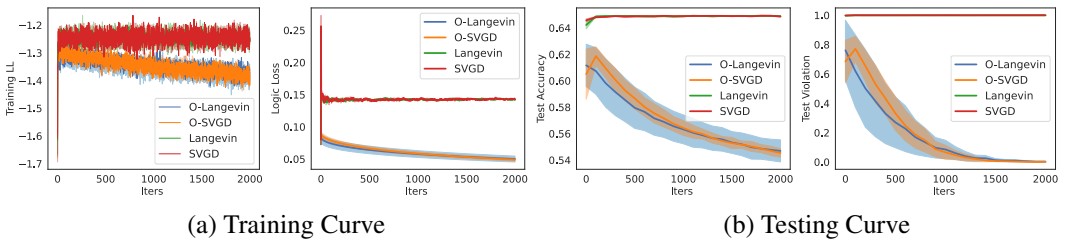

(a) Training Curve                 (b) Testing Curve

Figure 4: Loan Classification: O-Langevin and O-SVGD satisfy the constraint well whereas standard Langevin and SVGD violate the constraint severely.

CV score is defined as $|p(\text{income } 50,000|\text{female}) - p(\text{income } 50,000|\text{male})|$. Training and testing performance of our methods and standard algorithms are shown in Figure 3. We find that during training and testing, both O-Langevin and O-SVGD satisfy the constraint very well while maintaining high training LL and test accuracy. In contrast, standard Langevin and SVGD violate the constraint significantly, indicating that they fail to make fair predictions. These results demonstrate the power of O-Gradient which keeps the sampler on the manifold while still correctly sampling from $\pi(x)$.

**Loan Classification with Logic Rules**    Following previous work [25], we predict whether to lend loans to applicants and encode two logic rules: (1) an applicant must be rejected if not employed and having the lowest credit rank; (2) an applicant must be approved if employed over 15 years and having the highest credit rank. We use a Bayesian logistic regression model and set $g(x) = l_{\text{logic}}(x) = \mathbb{E}_{(\mu,\nu)\sim\mathcal{D}_{\text{logic}}}[\text{Loss}(\nu, \hat{\nu}(\mu; x))]$ where $\mathcal{D}_{\text{logic}}$ is a uniform distribution over all datapoints $(\mu, \nu)$ that satisfy the constraint, Loss refers to the classification loss and $\hat{\nu}$ is the prediction made by the model. During testing, we compute the violation which is the percentage of predictions that do not follow the logic rules. In Figure 4, we see that both of our methods satisfy the logic rules well whereas standard algorithms violate the constraint severely. A drop in the accuracy of our methods might be caused by conflicts between the data and the logic rules, that is, the solution with high accuracy does not satisfy the constraint. A similar phenomenon has also been observed in Liu et al. [25] which implies that this negative impact on the accuracy is most likely due to the task rather than the algorithm. Nevertheless, our methods can obtain models satisfying the constraint, thus guarantee their safety and interpretability.

**Prior-Agnostic Bayesian Neural Networks**    Bayesian neural network (BNN) has been widely used in deep learning (DL) for quantifying uncertainty. The posterior of BNNs is determined by the prior which reflects our prior knowledge, and the likelihood which quantifies the data fitness. Due to complex NN architectures, specifying an appropriate prior for BNNs has been shown to be difficult [8]. A poor prior often leads to a bad posterior and thus bad inference results. To automatically control the influence of the prior, one approach is to sample from the posterior with the constraint of a reasonably high data fitness. We apply our methods to image classification on CIFAR10 with ResNet-18. Since the training loss is typically very small in DL, we set the constraint to be $g(x) = \text{Loss}(x)$. Similar to stochastic Langevin dynamics (SGLD) [36] and SVGD, our methods are easy to be combined with stochastic gradients which are used on this task. We also ignore the second-order derivative terms in our methods for speedup. Besides standard algorithms, we also compare with a tempered

Table 1: Results on Image Classification (%). O-Langevin and O-SVGD significantly outperform unconstrained methods in terms of both generalization accuracy and calibration.

|  | Test Error ($\downarrow$) | ECE ($\downarrow$) | AUROC ($\uparrow$) |
|---|---|---|---|
| SGLD | 15.00 | 2.21 | 89.41 |
| Tempered SGLD | 4.73 | 0.83 | 97.63 |
| O-Langevin | **4.46** | 0.87 | **98.68** |
| SVGD | 6.11 | 0.93 | 93.55 |
| O-SVGD | 4.92 | **0.77** | 94.69 |

SGLD which has been used in the literature for good performance [38]. We report test error, expected calibration error (ECE) and AUROC on the SVHN dataset (measuring out-of-distribution detection).

From Table 1, we see that O-Langevin and O-SVGD improve significantly over their unconstrained counterparts on all three metrics. O-Langevin has the lowest test error and the highest AUROC while O-SVGD has the lowest ECE. These results indicate that the constraint can automatically limit the effect of the prior, leading to a better posterior and thus better generalization and calibration.

## 7 Conclusion and Limitations

We propose a new variational framework with a designed gradient flow (O-Gradient) for sampling in implicitly defined constrained domains. O-Gradient is formed by two orthogonal directions where one drives the sampler towards the domain and the other explores the domain by decreasing a KL divergence. We prove the convergence of O-Gradient and apply the framework to both Langevin dynamics and SVGD. We empirically demonstrate the power of our methods on a variety of ML tasks.

While ML achieves impressive performance, we must take realistic constraints into consideration when deploying it in daily life. This work provides a principled framework for solving constrained sampling problem with theoretical guarantees and practical algorithms.

One possible limitation of our methods is that the obtained samples are not exactly on the manifold due to working in an ambient space, though can be made arbitrarily close to the manifold by tuning hyperapameters. This might be solved by including a projection step. Another limitation is that we only consider an equality constraint. It will be interesting to incorporating inequality constraints into the framework. Besides, our theory only considers the convergence in the continuous-time limit. Future work could provide convergence analysis in discrete time, which may characterize the properties of practical algorithms more accurately.

## Acknowledgements

QL is supported by CAREER-1846421, SenSE2037267, EAGER-2041327, Office of Navy Research, and NSF AI Institute for Foundations of Machine Learning (IFML). XT is supported by the Singapore Ministry of Education (MOE) grant R-146-000-292-114.

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
