# OpenReview forum: "Sampling in Constrained Domains with Orthogonal-Space Variational Gradient Descent"
_NeurIPS.cc/2022/Conference — NeurIPS 2022 Accept_

### Official Review · Reviewer_nfnT · 2022-07-10

**Rating:** 6
**Confidence:** 4
**Soundness:** 2 fair
**Presentation:** 2 fair
**Contribution:** 3 good

**Summary:**

This paper introduced a sampling method, namely O-Gradient, that enables defining equality constraints which is achieved by two-component strategy: decrease the distance to the manifold by decreasing the KL divergence as fast as possible (using a velocity field. The paper presents two formulations using Langevin dynamics (O-Langevin) and SVGD (O-SVGD). The paper presents both theoretical analysis of the convergence of O-Gradient based on Proposition 5.2 that defines a limit that is used to formulate an orthogonal-space Stein equation. This in turn is used to generalize the Fisher divergence. The experimental evaluation considered both synthetic and real scenarios. In the former, the paper presents analysis of convergence comparing O-Gradient with  constrained Langevin dynamics and constrained Hamiltonian Monte Carlo, as baselines, where O-Gradient methods show better convergence when sampling from inside or outside the manifold. The baseline methods have a MAE is almost zero for when sampling is done from inside and are not applicable for sampling from outside the manifold. The real-world experiments are done for income classification, loan classification, and image classification where O-Gradient methods seem to perform better than the alternatives.

**Questions:**

Could you clarify why the FPE of the definition of $dx_t$ in line 150 does not follow Eq (10).
What function is used to compute the CV score in the Income Classification with Fairness Constraint ?
What is the function used to compute the error in Loan Classification with Logic Rules?
What is the function that was used to compute the test error in the experiment “Prior-Agnostic Bayesian Neural Networks”?

**Limitations:**

The paper describes the limitations in terms of the nature of the model (i.e., it is an approximation and samples are not exactly on the manifold, the type of constraint is equality only, and continuous time).
No limitations are provided outside this technical realm.

**Strengths And Weaknesses:**

In summary, the paper is somewhat clear but more could be added to make it self-contained. The organization is not bad. The originality is mostly about the specific equality constrain considered during sampling. The quality of the work is theoretically good and experimentally reasonable. Some specific details include:

Strengths:
Variational gradient descent applied to sampling with equality constraints is relatively novel problem. It is related to, but not the same than, variational inference subject to inequality constraints for which there are works that propose a solution.
It is an interesting approach to try to define sampling in terms of closeness and orthogonality to the related manifold defined by the domain of the distribution and the equality constraint.

Weaknesses:
The paper, as written, is only accessible to the ML sub-community working on Sten variational inference and sampling and related topics as it does not present the intuition of the work but rather in general terms. For instance, the paper states that Fokker–Plank equation of the SDE $dx_t$ (dubbed by the paper as natural guess) does not follow the density flow expected in Eq 10. Further details could clarify this.

There are other sections of the paper where the explanations provided are insufficient such as the lack of a data description in the experiment for Loan Classification with Logic Rules. The function used to measure the test error in the experiment “Prior-Agnostic Bayesian Neural Networks” is not detailed.

---

> ### Author Response · Authors · 2022-08-02
> **Authors' Response to Reviewer nfnT**
>
> Thank you for your thoughtful review. We answer your questions below.
>
>
> **Q: The paper is only accessible to a ML sub-community and does not present the intuition.**
>
>
> A: We have indeed provided extensive intuitions about our method. In Figure 1, we have visualized the main key ideas of O-Gradient and the two resulting algorithms, showing how the gradient is decomposed to achieve constraint sampling. When introducing our variational framework in Section 4.2, we have mentioned in L117 (in the updated version) that the meaning of the objective is to maximize the decay of $KL(q \|| \pi)$, and also ensure q to get towards the manifold. When introducing our practical algorithms in Section 4.3, we have provided intuitions of the construction in L137, where our algorithms keep the original SVGD or Langevin dynamics movement in the directions perpendicular to $\nabla g$, while pushing the density towards $G_0$ along the $\nabla g$ direction.
>
>
> This paper has technical depth since we developed a theoretically grounded framework to achieve efficient constraint sampling. By its nature, the technical details are unavoidable and it is in general impossible to provide extensive background knowledge in a nine-page paper. We will try our best to offer more intuitions in the revision. But we believe that its technical depth is an advantage rather than the reason for rejection.
>
>
> **Q: Fokker–Plank equation of the SDE dxt (dubbed by the paper as a natural guess) does not follow the density flow expected in Eq 10.**
>
>
> A: First of all, we introduce the naive projected SDE as a stepping stone to derive the correct solution. Its FPE is not of theoretical importance. Given your response, we think discussing this naive SDE may confuse other audiences as well and decide to move its presence to the appendix (see the remark in the proof of Theorem 4.2).
>
>
> To answer your question about why this naive SDE does not have the correct FPE, we can write its FPE explicitly
>
> $\frac{d}{dt}q=-\nabla \cdot (\phi(x) q(x))+\sum_{i,j} \partial_{i,j}^2 (D_{i,j} (x)q(x))$
>
> which clearly shows that it misses a term and is not equivalent to Eq 10.
>
>
>
> **Q: Data description in the experiment for Loan Classification with Logic Rules.**
>
>
> A: We have described the dataset in Section B.3 in Appendix. We copied it below.
>
> The dataset\footnote{https://www.kaggle.com/datasets/adarshsng/lending-club-loan-data-csv} contains loans issued through 2007-2015 of several banks. Each data point contains 28 features such as the current loan status and latest payment information. We define the logic loss to be the binary cross-entropy loss. The metric values are the mean over all particles. For both methods, we use $n=10$ and $\beta=0$ . For O-Langevin, $\alpha=80$ and $\eta=10^{-4}$. For O-SVGD, $\alpha=100$ and $\eta=10^{-3}$. The results are averaged over 3 runs with the standard error as the error bar.
>
>
>
>
> **Q: Metrics used in the experiments.**
>
>
> A: CV score is a standard fairness measure of disparate impact [1,2]. It is defined as |p(income$\ge$ 50,000|female) - p(income$\ge$ 50,000|male)|.
>
>
> In Loan Classification with Logic Rules, the test violation refers to the percentage of predictions that do not follow the logic rules.
>
>
> In Prior-Agnostic Bayesian Neural Networks, the test error refers to the percentage of the wrong predictions on the test set.
>
>
> All the metrics in the experiments are standard and widely used in the literature. We have added these details in the appendix.
>
>
>
> [1] Three naive bayes approaches for discrimination-free classification, Data Mining and Knowledge Discovery 2010
>
> [2] Fairness-aware Classifier with Prejudice Remover Regularizer. In PADM, 2011.

---

> > ### Comment · Reviewer_nfnT · 2022-08-08
> > **Response**
> >
> > Thank you for answering my questions. This clarifies most of my doubts about the contribution of this paper. I suggest including the information in your reply in the main text of the paper. I will raise my score in consequence.

---

> > > ### Author Response · Authors · 2022-08-09
> > > **Thank you**
> > >
> > > Thank you for your reply and for raising your score. We will be sure to include the above information in the main text of the revision.

---

### Official Review · Reviewer_K6fK · 2022-07-10

**Rating:** 7
**Confidence:** 4
**Soundness:** 1 poor
**Presentation:** 3 good
**Contribution:** 3 good

**Summary:**

The paper introduces a class of algorithms for sampling from a target density conditionally on a set of constraints expressed as an equation $g(x)=0$. The paper relies on a variational formulation of the vector fields defining Langevin dynamics and SVGD to derive an Orthogonal- version of such vector field with two components: an orthogonal component that drives samples toward the set of constraints and a tangential component that diffuses the particles to that they are distributed according to the target. The algorithm is simple to compute and requires only evaluating the score of the target, the constraint g(x), and its gradient.
Convergence rates for the algorithm are provided and rely on a new stein characterization of conditional measures which appears to generalize stein characterization.

**Questions:**

Proof of prop 5.4:
        - It appears the authors apply the k-PI inequality of Definition 5.3 to the function f(y) = \sqrt{q_z(y)/Pi_z(y)}. When computing the variance of f under $\Pi_z$, I get the expression: $1- a^2$ with a = E_{Pi_z}[f], so that k-Pi would be given by $(1-a^2)\leq E[||D\nabla ||^2]$. However, in the proof, the authors use the inequality $2(1-a)\leq E[||D\nabla ||^2]$. I do not see why this would hold, since $1-a^2 \leq 2(1-a) $.
       - The equation after line 493 suggests that the score difference between $q_z$ and $\Pi_z$ is the limit of the scores $\pi_{\eta,z}$, but I do not see why this is true. It seems this claim follows from Proposition5.2. However, the latter only shows that the densities $\pi_{\eta,z}$ converge towards Pi_z in the 'weak limit'. This result does not necessarily imply that scores of $\pi_{\eta,z}$ converge to the score of $PI_z$ in a pointwise sense. Please clarify this.

- Statement of thm 5.5: I think the statement should be modified as follows:

Set $M_t= sup(|g(x)|, x\in supp (q_0)  )$ and define S_t by dS_t = -psi(S_t) and $S_0=M_0$. If M_0<+infty, then M_t\leq S_t for all times.  Currently, the paper makes the stronger claim that dM_t=-\psi(M_t), but this is not proven and is not really needed for the rest of the paper.

In eq 17, there is a factor 2 missing to the 3 terms otherwise the proof in the appendix doesn’t hold.

Following the proof of thm 5.6, it seems that the conclusion of thm 5.6 on the upper bound of the O-stein discrepancy is wrong. Instead, the upper bound should hold only for the min_t over [0,T] and not for any t in [0,T].

Eq $17$ does not seem to be an appropriate smoothness assumption: when g is smoother the l.h.s in 17 blows up to infinity, so that the constant C_0 must be larger and larger. In the limit case when g is constant, the inequality never holds. This can be fixed by scaling the l.h.s by the norm of $||g||$ and replacing the last term in L 228 by the ratio (\psi(g(x))/\Vert g(x)\Vert).




**Limitations:**

The authors should discuss how restrictive are the assumptions required for the results + fix the proofs. I am willing to raise my score to a (weak) acceptance it these issues are addressed in an acceptable manner.

**Strengths And Weaknesses:**

The idea appears to be novel and might have a significant impact in the context of conditional sampling under generic equation constraints. The algorithms are simple to use and appear to be effective on the considered examples.
The presentation is also clear and the problem well motivated.


My main concern is about the soundness of the theory as it appears there are some errors or unclear steps in the proofs (see questions section for more details). I hope the authors can fix these issues and clarify these points. More generally, the write-up of the proofs should be improved to better explain the intermediate steps in the calculations and add explicit logical links between different components of the proofs.

It seems that the main result relies on at least two strong assumptions: the Poincaré inequality for conditional densities and the inequality in eq (17) on the constraint $g$. Can the authors provide examples where such assumptions hold? Classical PI holds in many situations, but for the conditional case, it is still unclear to me how restrictive this is.

---

> ### Author Response · Authors · 2022-08-02
> **Authors' Response to Reviewer K6fK**
>
> Thank you for your constructive review. We answer your questions below.
>
> **Q: The Poincaré inequality for conditional densities.**
>
> A: PI on Riemannian manifold has been studied before. In particular, the manifold is compact and the measure has a bounded and smooth density, PI holds with a constant that may depend on the radius of the manifold. For many applications, $g(x)=0$ can be compact or we may be only interested in distributions with compact support. For non-compact manifolds, there are simple examples where $g$ is affine, $\pi$ is Gaussian, so $\Pi_z$ is also Gaussian. But for general manifolds, of course this is a very difficult question, because the notion of convex functions is defined only for convex domains.
>
> **Q: Inequality in eq (17).**
>
> A: We have fixed the missing coefficients and updated the formulation using your suggestions, thank you! Now we only need $g$ to be smooth and $\|\nabla g\|$ to be bounded when $g$ is bounded. These are much less restrictive than the earlier version.
>
> **Q: Proof of prop 5.4.**
>
> A: You are right that we missed a few steps here. With your notation, we note that $a\le1$, so $1-a\le1-a^2\le E[\|D \nabla \|^2]$. So a similar result can be obtained.
> Your comment on the weak limit is also correct. We have added a new Lemma A.1 and some new regularity requirements on the density to reach the claim.
>
> **Q: Statement of thm 5.5.**
>
> A: We have updated based on your suggestion.
>
>
> **Q: Statement of thm 5.6.**
>
> A: You are right. We have fixed it.

---

> > ### Comment · Reviewer_K6fK · 2022-08-08
> > **Thank you for your reply, I will raise my score**
> >
> > Thank you for the clarifications.
> > I am now more confident about the proofs. I have also read the other reviews and couldn't find, in my opinion, any major concern that should be an obstacle to accepting the paper at this conference. Therefore, I will raise my score to 7 since the idea is original and is used to solve a new class of sampling problems that might be of interest to the community.
> >
> >
> > Here are a few comments:
> > - I'm happy with lemma A.1, although the writeup could be technically a bit more accurate: perhaps defining the ratio as a Radon-Nykodim of q_z(x) wrt to \pi_z(x) and writing the proof without introducing the map t(z,y) (which requires justification for its existence).
> >
> >
> > - Regarding the PI, I understand the argument, however, when dealing with conditional densities, the PI constant might depend on $z$ and could blow up even though the manifold is compact (is the dependence of such constant on $z$ is discontinuous).

---

> > > ### Author Response · Authors · 2022-08-09
> > > **Thank you**
> > >
> > > We are glad that you are happy with our changes. Thank you for raising your score on us. Your further suggestions are also very helpful.
> > >
> > > Lemma A.1: We have implemented your suggestion by writing integration in a more measure theoretical manner. They have made the proof more concise and elegant. In particular, the transformation $t$ is no longer needed.
> > >
> > > You are right. The proof in [11] only shows that $\kappa$ exists, but not how big it is. So technically we can’t guarantee that a uniform $\kappa$ exists for $\Pi_z$ with a range of $z$ (maybe with further study we can, but that is a more math theory question, and it will not be the focus of this paper). We have added a line “But it is often hard to find the exact PI constant $\kappa$ for $\Pi_z$, especially if we are interested in a family of $\Pi_z$ with possibly discontinuous dependence on $z$.”

---

### Official Review · Reviewer_aCms · 2022-07-11

**Rating:** 6
**Confidence:** 2
**Soundness:** 3 good
**Presentation:** 3 good
**Contribution:** 3 good

**Summary:**

This paper proposed a new variational framework for sampling from constrained domains. By designing an orthogonal-space gradient flow, i.e. the O-Gradient, the proposed methods can sample from manifolds defined by general equality constraints. The O-Gradient can be realized by both the Langevin dynamics and the Stein variational gradient descent, namely the O-Langevin and the O-SVGD. In contrast to previous methods, the O-Langevin and the O-SVGD do not require initializing the samples on the manifolds defined by the constraints. Besides, the paper has proved that the O-Gradient converges to the target distributions with rate $\tilde{\mathcal{O}}(1/T)$, where T is the number of iterations. In the experiment section, the O-SVGD and the O-Langevin have shown their effectiveness in various setups, showing that they can generate samples that approximate the target distribution well while also staying on the manifold defined by the constraints.

**Questions:**

1. In Section 4.1, "$\frac{dq}{d\pi}$ and $KL(q||\pi)$ are ill-posed", what is the definition of "ill-posed"?
2. What's the interpretation of Eq. (5)? Why do you choose the function $\psi(x)=\alpha\text{sign}(x)|x|^{1+\beta}$?
3. In line 149, what is the "projection intuition"?
4. Line 252, what is the "energy distance"?

**Limitations:**

Yes, the author has adequately addressed the limitations and potential negative societal impact of their work. The limitations of the approach were presented above.

**Strengths And Weaknesses:**

Strength:
1. The paper has addressed an interesting topic of variational samplers.
2. The proposed methods have made remarkable progress upon previous methods, alleviating the requirement of initializing samples on the targeted manifold.

Weakness:
1. It seems that the proposed method has some negative impact on the original task, e.g. in the income classification experiment, a decrease in the test accuracy can be observed.
2. It is unclear what the computational cost of this method is compared to the compared baselines.

---

> ### Author Response · Authors · 2022-08-02
> **Authors' Response to Reviewer aCms**
>
> Thanks for your supportive and thoughtful comments. We answer your questions below.
>
> **Q: The proposed method has some negative impact on the original task.**
>
> A: As stated in L295 (in the updated version), this phenomenon is caused by the conflict between the data and the constraint, that is, the solution with high accuracy does not satisfy the constraint.  This is verified by the results in Figures 3&4, where the solutions from the standard Langevin and SVGD, though have high accuracy, violate the constraints severely. A similar phenomenon has also been observed in previous work [4] which implies that this negative impact is most likely due to the task rather than the algorithm itself.
>
> **Q: It is unclear what the computational cost of this method is compared to the compared baselines.**
>
> A: To the best of our knowledge, our method is the first constraint sampling without the requirement of initializations on the manifold, so there is essentially no baseline that can achieve the same effect. Compared to the unconstrained Langevin and SVGD, our method additionally computes the gradient and the Hessian of the constraint function. Compared to previous manifold sampling methods which require expensive projection subroutines, our method has a much cheaper and faster update. For example, in the synthetic distribution experiment, one update of O-Langevin (ours) takes 0.023s whereas the previous method CLangevin takes 0.08s. From Figure 2a, we can see that O-Langevin also converges faster than CLangevin in terms of the number of iterations.
>
> **Q: what is the definition of "ill-posed"?**
>
> A: We have rephrased it as _$q$ is singular w.r.t. $\pi$_, so both $\frac{dq}{d\pi}$ and $KL(q\||\pi)$ _are not defined_.
>
> **Q: What's the interpretation of Eq. (5)? Why do you choose the function?**
>
> A: The first term maximizes the decay of $KL(q || \pi)$ as in unconstraint sampling, and the second term makes sure that the velocity field $v_t$ drives q towards the manifold $g(x)=0$. To see the second term effect, we consider three cases:
>
> 1) When $g(x) > 0$, then $v_t(x)^\top\nabla g(x)=-\psi(g(x))>0$ which ensures that $g$ decreases strictly.
>
> 2) When $g(x) < 0$, then $v_t(x)^\top\nabla g(x)=-\psi(g(x))<0$ which ensures that $g$ increases strictly.
>
> 3) When $g(x) = 0$, then $v_t(x)^\top\nabla g(x)=-\psi(g(x))=0$ which ensures $x$ to stay on the manifold $g(x)=0$.
>
> We choose $\psi(x)=\alpha \text{sign}(x)|x|^{1+\beta}$ because it is one of the simplest functions that satisfy the requirements and we found it works well in theory and practice. We will add the above explanation in the revision.
>
> **Q: In line 149, what is the "projection intuition"?**
>
> A: The intuition is more of a naive guess: we simply project the drift and dW_t in the Langevin dynamics and see if the corresponding SDE is already the answer. In the hindsight, this seems not intuitive and also confused Reviewer nfnT, as Reviewer nfnT thinks this is a very important issue/drawback (which is not). So we decided to remove it from the main body and discuss this naive guess in the appendix (see the remark in the proof of Theorem 4.2).
>
> **Q: Line 252, what is the "energy distance"?**
>
> A: Energy distance is a statistical distance between probability distributions and has been used in the literature, e.g. [1, 2, 3]. We used it to measure the difference between the approximated distributions by sampling methods and the target distribution. Formally speaking, the energy distance between probability distributions P and Q is defined by
> $D(P,Q) = 2E_{Z,W}\||Z-W\||_2-E_{Z,Z}\||Z-Z'\||_2-E_{W,W'}\||W-W'\||_2$
>
> where $Z,Z'\sim P$ and $W,W'\sim Q$. We have added the above details in the revision.
>
> [1] Equivalence of distance-based and RKHS-based statistics in hypothesis testing, Annals of Statistics 2013
>
> [2] Hypothesis testing using pairwise distances and associated kernels, ICML 2012
>
> [3] A Spectral Energy Distance for Parallel Speech Synthesis, NeurIPS 2020
>
> [4] Sampling with Trusthworthy Constraints: A Variational Gradient Framework, NeurIPS 2021

---

### Official Review · Reviewer_p8UL · 2022-07-12

**Rating:** 5
**Confidence:** 2
**Soundness:** 3 good
**Presentation:** 3 good
**Contribution:** 3 good

**Summary:**

The paper explores an orthogonal space gradient descent. This is achieved by constructing the perpendicular space from the gradient direction using $\mathbb{I} -\frac {\nabla g \nabla g^T}{\norm{g}^2} $

**Questions:**

I do not have sufficient context on the paper, therefore I do not have any questions at this point.

**Limitations:**

Yes, the authors have addressed the impact of their work sufficiently.

**Strengths And Weaknesses:**

The reviewers expertise is limited in this field, hence the comments will be very brief and editorial.
Strengths:
1. The paper is well written. Both the language and the math seem to be consistent.


Weaknesses:
1. The applications might not exactly be suitable for a conference like Neurips
2. The complexity is of the algorithm is not mentioned.

---

> ### Author Response · Authors · 2022-08-02
> **Authors' Response to Reviewer p8UL**
>
> Thank you for your valuable review. We answer your questions below.
>
> **Q: The applications might not exactly be suitable for a conference like Neurips.**
>
> A: We would like to emphasize that constraint sampling is an important and suitable topic for machine learning conferences. Taking last year's NeurIPS as an example, several papers are on this topic, such as [1, 2]. Constrained sampling is a very needed technique for many critical areas, such as enforcing trustworthy constraints in ML systems.  More importantly, this paper makes significant contributions to the field, by providing new theoretical analysis from a variational view and two practical algorithms. Both our theory and algorithms remove the strong assumptions of initializing and keeping the sample on the manifold. Therefore, we believe this paper is a suitable and valuable contribution to NeurIPS.
>
>
> **Q: The complexity of the algorithm is not mentioned.**
>
> A: We have discussed the asymptotic complexity in Theorem 5.6, which shows that our method converges to the target constrained distribution with rate $\widetilde{O}(1/\text{the number of iterations})$. Regarding the cost per step, the main cost of our method is computing the gradient and the Hessian. Our method has a much cheaper and faster update per step, compared to previous manifold sampling methods which require expensive projection subroutines, such as Newton’s method or a long-time ordinary differential equation.
>
> In practice, we also found that our algorithms have a cheaper update and a faster convergence. For example, in the synthetic distribution experiment, one update of O-Langevin (ours) takes 0.02s whereas the previous method CLangevin takes 0.08s. From Figure 2a, we can see that O-Langevin converges faster than CLangevin in terms of the number of iterations.
>
> [1] Efficient constrained sampling via the mirror-Langevin algorithm, NeurIPS 2021
>
> [2] Sampling with Trusthworthy Constraints: A Variational Gradient Framework, NeurIPS 2021

---

### Meta-Review · Area_Chair_pHRx · 2022-08-24

**Recommendation:** Accept
**Confidence:** Less certain

**Metareview:**

All reviewers recommend accepting the paper, to various levels of enthusiasm.

When preparing the final version, please take into the following considerations:
- Several of the reviewers pointed out that the paper was unclear/sloppy in places, and that it was not written in a way that is accessible to a general ML audience. Take some time to fix this; people are more likely to read/appreciate/cite/build-on your work if it written in an accessible way, with clear motivation (understandable beyond people in a subfield), and the steps in the analysis are laid clearly in an easy-to-understand way.

**Award:**

No

---

### Decision · Program_Chairs · 2022-09-14

Accept